# Assessment of MRI-Linac Economics under the RO-APM

**DOI:** 10.3390/jcm10204706

**Published:** 2021-10-14

**Authors:** Russell F. Palm, Kurt G. Eicher, Austin J. Sim, Susan Peneguy, Stephen A. Rosenberg, Stuart Wasserman, Peter A. S. Johnstone

**Affiliations:** 1Moffitt Cancer Center, Department of Radiation Oncology, 12902 USF Magnolia Drive, Tampa, FL 33612, USA; austin.sim@moffitt.org (A.J.S.); Susan.Peneguy@moffitt.org (S.P.); stephen.rosenberg@moffitt.org (S.A.R.); stuart.wasserman@moffitt.org (S.W.); peter.johnstone@moffitt.org (P.A.S.J.); 2Guidehouse Healthcare Consulting, 150 North Riverside Plaza, Suite 2100, Chicago, IL 60606, USA; kurt.eicher@guidehouse.com

**Keywords:** MRI-Linac, RO-APM, adaptive radiotherapy, MRgART, medicare, reimbursement, radiation oncology

## Abstract

The implementation of the radiation oncology alternative payment model (RO-APM) has raised concerns regarding the development of MRI-guided adaptive radiotherapy (MRgART). We sought to compare technical fee reimbursement under Fee-For-Service (FFS) to the proposed RO-APM for a typical MRI-Linac (MRL) patient load and distribution of 200 patients. In an exploratory aim, a modifier was added to the RO-APM (mRO-APM) to account for the resources necessary to provide this care. Traditional Medicare FFS reimbursement rates were compared to the diagnosis-based reimbursement in the RO-APM. Reimbursement for all selected diagnoses were lower in the RO-APM compared to FFS, with the largest differences in the adaptive treatments for lung cancer (−89%) and pancreatic cancer (−83%). The total annual reimbursement discrepancy amounted to −78%. Without implementation of adaptive replanning there was no difference in reimbursement in breast, colorectal and prostate cancer between RO-APM and mRO-APM. Accommodating online adaptive treatments in the mRO-APM would result in a reimbursement difference from the FFS model of −47% for lung cancer and −46% for pancreatic cancer, mitigating the overall annual reimbursement difference to −54%. Even with adjustment, the implementation of MRgART as a new treatment strategy is susceptible under the RO-APM.

## 1. Introduction

In September 2020, the Centers for Medicare and Medicaid Services (CMS) submitted the revision to a new bundled reimbursement model for radiation oncology services provided to Medicare patients with a planned implementation date of 1 January 2022 [1]. Initially mandatory for 40% of providers, the Radiation Oncology Alternative Payment Model (RO-APM) will cover 16 different cancer types and will simplify reimbursement to bundled payments delivered twice over a 90-day period of service. The rationale behind this change is multifactorial; however, a fundamental aim is to separate from the traditional fee-for-service (FFS) model that generally rewards volume over value. Some aspects of the proposed payment model are incompletely defined, such as the historical payment experience, case mix and efficiency [2]. The latter is meant to reduce payment for higher cost practices but does not account for situations where higher cost care may be the most appropriate modality of treatment.

A concern arising from such a new payment model is that a lower reimbursement may restrict oncology practices from making investments in new technology thereby suppressing innovations in care. A promising technology that will be impacted by these changes is the magnetic resonance guided linear accelerator (MRL), which allows for radiation treatment delivery with real-time image guidance as well as daily treatment adaptation for dynamic changes in patient anatomy. The difficulty in utilizing such online adaptive planning is that it is time-intensive and requires physician and physicist staffing costs that traditional linear accelerators do not require. As novel treatments and techniques tend to be more costly until efficiency is optimized, national committee (American Society for Radiation Oncology; ASTRO) recommendations have asked for a stay on including this new technology into the RO-APM [3]. In this manuscript, we sought to explore the impact of the RO-APM on technical billing reimbursement and to determine a reimbursement strategy, whereby MR-guided adaptive radiotherapy (MRgART) may be viable.

## 2. Materials and Methods

All models utilized an annual MRL patient load and distribution consisting of 200 patients with 5 primary cancer diagnoses: breast (31%), lung (13%), rectal (15%), pancreas/hepatobiliary (28%) and prostate (13%). This patient mix was derived from an internal model as well as published literature [4]. Online adaptive treatment planning was used in both lung and pancreatic cancer and MRI guidance without adaptive planning was used for breast, colorectal and prostate cancer. All treatment regimens except rectal cancer (28 fractions) were hypofractionated in concordance with trends in modern practice and to emphasize efficiency of care: breast: 5 fractions [5], lung: 8 fractions [6,7], prostate: 5 fractions [8,9,10], pancreas: 5 fractions [11,12].

The procedural terminology (CPT) codes (Appendix A) and frequency of billing for the technical components of treatment were tabulated with the current capabilities of the MRL consisting of step-and-shoot intensity modulated radiation therapy (IMRT). The “scalable” CPT codes that account for the technical billing for treatment adaptation consisted of 77300, 77301 and 77338 and use may be justified with dosimetric evidence with an original treatment plan that would lead to an overdose of organs at risk or offered inferior target coverage compared to the new daily adapted plan. As the RO-APM currently only applies to Medicare patients, we utilized nationwide average traditional Medicare FFS reimbursement rates for the CPT codes associated with technical fees and compared these to the bundled reimbursement with the diagnosis in RO-APM. The mechanics, as well as site neutral calculations for the RO-APM model, have been previously published [2,13].

Our second exploratory aim was to formulate a modified repayment strategy within RO-APM (mRO-APM) that accounted for the necessary resources for replanning per case. We proposed that the base rate of the modifier be the full reimbursement for the adaptive treatments under the current FFS model as delivered by modern volumetric modulated arc therapy (VMAT) technique with 2-arc plans. This base rate modifier was added to the subtotal of the RO-APM after geographic adjustment but may be conservatively scaled by the proportion of treatments requiring daily replanning (≤20%: ×0, 21–79%: ×0.25, ≥80%: ×0.5). For this analysis, all lung and pancreas patient treatments were assumed to have all fractions adapted.

## 3. Results

Annual and per case reimbursement for all models are reported in Table 1. Per case reimbursement for all selected diagnoses were lower in the RO-APM compared to FFS with large differences in the adaptive treatments for lung cancer (−89%) and pancreatic cancer (−83%) and more moderate differences for nonadaptive treatments (breast: −74%; colorectal: −65%; prostate: −48%). The total annual reimbursement discrepancy in the RO-APM compared with FFS was −78%.

Without implementation of adaptive replanning there was no difference in reimbursement in breast, colorectal and prostate cancer between RO-APM and mRO-APM. Accommodating online adaptive treatments in mRO-APM would result in differences of −47% for lung cancer and −46% for pancreatic cancer with an overall annual difference of −54% from the FFS model. The planned 5-year implementation of the RO-APM in its current state would result in a deficit of 42.8 million dollars from the FFS model, which was reduced to a difference of 29.7 million dollars in the mRO-APM model (Figure 1).

## 4. Discussion

As we seek to improve oncological outcomes, radiation oncology has been directed to engage in a new APM to mitigate financial toxicity for patients [14,15] as well as the cost of the national cancer care burden which is estimated to reach USD 246 billion in 2030 [16]. In its current form, the RO-APM is treatment agnostic and techniques with higher overhead such as proton therapy [17,18] and MRgART [19,20] will be differentially impacted despite emerging potential benefits to patients. A similar report on adaptive treatment reimbursement under the RO-APM demonstrated lower but substantial decreases in reimbursement for pancreas (−36%), lung (−23%) and liver (−8%) malignancies and suggested that implementation of the APM would limit growth and innovation [21].

Historically, the widespread adoption of new technology without prospective evidence has been scrutinized. For instance, a SEER analysis noted that the rapid adoption of IMRT for prostate cancer from 2002 to 2005 resulted in increased spending in excess of USD 200 million dollars [22]. However, the majority of these treatments were likely in standard fractionation over 7–8 weeks and current standard of care has now shifted with the growing adoption of hypofractioned treatments [23]. These treatments carry greater value and are fundamentally reliant on the improved dose distributions derived from IMRT treatment plans. Furthermore, shorter treatment times associated with hypofractionation also decrease patient burden by mitigating travel and time away from work and family. We would predict that with ongoing development of the MRL, faster dose calculations and volume auto-segmentation [24,25] may help to lower costs and improve treatment value in manner similar to IMRT and SBRT.

A key proposed benefit of MRgART is the ability to account for inter- and intra-fraction motion during treatment and minimize or eliminate internal target volumes (ITV) with respiratory gating. Daily plan optimization has demonstrated stark dosimetric benefits to organs at risk (OARs) in thoracic or abdominal tumor locations where data demonstrate that upwards of 90–95% of treatments may benefit in adaptation by either reducing organ at risk dose or improving tumor coverage [26,27]. Specific diseases where clinical data are maturing for MRgART include pancreas [11,12,28,29], ultra-central thoracic malignancies [30,31], liver [32,33], prostate [34] and bladder cancer [35]. The use of on-board MRI imaging may also lead to other advances in radiomics and disease response assessment [36,37,38] which should improve patient selection and allow for risk-adapted treatment strategies. While daily image guidance is currently built into the technical billing reimbursement, these new advancements require significant translational support and alternative reimbursement strategies should be pursued.

Another key weakness of the RO-APM is that financial risk factors such as advanced age and disease stage are not considered and centers that care for these patients may be further disadvantaged by the RO-APM [39]. Furthermore, radiation oncology accounted for 1.4% of total Medicare charges in 2017 and is hundreds of millions of dollars lower than diagnostic radiology (3.8%) and medical oncology (6.6%) [40] for whom a specific driver of increased cost is metastatic disease [41,42]. In an analysis of commercial payments for breast, colorectal and lung cancer patients, radiation therapy and imaging accounted for less than 15% of the total cost of care while chemotherapy and supportive drug therapies accounted for 48% [42]. In the future, oligometastatic or oligoprogressive disease targeted radiation therapy may not only reduce need for prolonged chemotherapy for the patient but also may mitigate the associated financial toxicity of expensive or ongoing systemic treatments.

Finally, the era of bundled payments carries the benefit of streamlined compensation and should inherently encourage radiation oncologists to improve the value and efficiency in treatment. Analogous processes are ongoing in gynecological oncology in the treatment of low-risk endometrial cancer [43] and recommendations to risk-stratify bundled payments in orthopedic surgery have been published [44,45]. While radiation oncologists should strive to continue to be good financial stewards in the health care system, increased effort to improve the therapeutic index of the treatments provided should be fairly rewarded.

Limitations of this study include some currently undefined variables in the RO-APM including the trend factor, geographic adjustment, case mix adjustment and historical experience which in this study were left at neutral values [2]. However, in this analysis, predefined initial discounts and witholdings were included that may put practices with thin margins at risk of financial loss. While we used national median reimbursement data for FFS billing, significant regional variability exists and differentially impacts practitioners [46]. The geographic adjustment in RO-APM may help mitigate these discrepancies but is unlikely to fully ameliorate them. Finally, we sought to focus this analysis on the billing of technical service as it generally is more financially significant compared to billing for professional services [47]. Modifiers or adjustments on the professional bundled payments of the RO-APM for more time intensive or complex treatment planning would further help to appropriately compensate individual physician time when they are not generating revenue through traditional means.

## 5. Conclusions

Investment into new technologies such as MRgART will be cost-prohibitive to most American oncology care providers under the RO-APM. We propose that modifying the RO-APM to reimburse for adaptive treatment planning and delivery as special treatment procedures would allow for continued growth and innovation in oncology care.

## Figures and Tables

**Figure 1 jcm-10-04706-f001:**
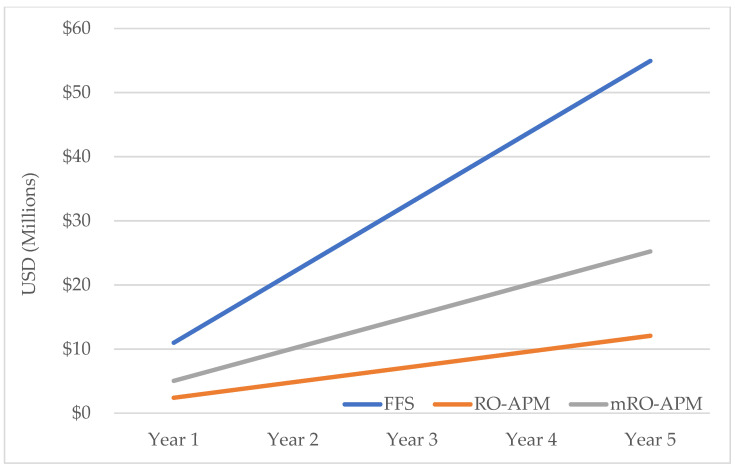
Projected annual reimbursement in millions USD across 5 years under each reimbursement model.

**Table 1 jcm-10-04706-t001:** Per case reimbursement for selected cancer diagnoses under each reimbursement model and projected annual reimbursement based on model patient population. Abbreviations: fee-for-service, FFS; Radiation Oncology Alterative Payment Model, RO-APM; m, modified.

Model	Breast	Lung	Colorectal	Prostate	Pancreas	Projected Annual Total
FFS	USD 36,576	USD 102,953	USD 32,284	USD 36,576	USD 73,669	USD 10,989,441
RO-APM	USD 9493	USD 11,241	USD 11,284	USD 18,978	USD 12,544	USD 2,415,309
mRO-APM	-	USD 54,357	-	-	USD 39,492	USD 5,045,390

## Data Availability

Publicly available datasets were analyzed in this study. This data can be found here: https://www.cms.gov/Research-Statistics-Data-and-Systems/Files-for-Order/LimitedDataSets/MEDPARLDSHospitalNational.

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
