# Peer review of "Assessment of MRI-Linac Economics under the RO-APM"

_jcm, 2021, doi:10.3390/jcm10204706_

Round 1
Reviewer 1 Report
Interesting paper exploring the impact of the RO-APM on technical billing reimbursement and to determine a reimbursement strategy whereby MR-guided adaptive radiotherapy (MRgART) may be viable.
I appreciate the balance with limits
Reimbursement system in the era of advanced radiotherapy claim to be updated, in my opinion this is an unmeet need that international radiotherapy societies have to face otherwise our strength in modern radiotherapy delivery, precision of treatment planning and delivery, QA, and alternative schedule are not well valorized in several national reimbursement system
Reviewer 2 Report
The authors present a concise evaluation of the impact of the RO-APM on MRI-guided adaptive therapy and make a compelling case that reduced reimbursements within this model are a distinct disincentive for investment, use, and further development of this potentially paradigm shifting technology. The authors provide a brief discussion of the importance of innovation and necessary time for promising new technologies to mature, optimize efficiency, and provide maximal value. A reasonable comparison is provided between RO-APM payments for a standard practice with MR-linac and an alternative model where fee-for-service payments are retained at current VMAT levels for MRgART delivery. This comparison highlights the dramatic reductions in payments for adaptive treatments using MRgART under the RO-APM, over-and-above the reductions predicted for non-adaptive treatments. This is a timely evaluation and will support and motivate additional more detailed investigations of the economics of MRgART implementation. Additional analysis to detail the expenses associated with MR linac practice, as well as downstream financial and QOL benefits of improved quality of care afforded by MRgART, will be of particular interest, but beyond the scope of this concise evaluation.